# High-Throughput In Vitro Screening Identified Nemadipine as a Novel Suppressor of Embryo Implantation

**DOI:** 10.3390/ijms23095073

**Published:** 2022-05-03

**Authors:** Xian Chen, Sudini Ranshaya Fernando, Yin-Lau Lee, William Shu-Biu Yeung, Ernest Hung-Yu Ng, Raymond Hang-Wun Li, Kai-Fai Lee

**Affiliations:** 1Department of Obstetrics and Gynaecology, LKS Faculty of Medicine, The University of Hong Kong, Hong Kong, China; chenxianchristine@163.com (X.C.); sudini@uwu.ac.lk (S.R.F.); cherielee@hku.hk (Y.-L.L.); wsbyeung@hku.hk (W.S.-B.Y.); nghye@hku.hk (E.H.-Y.N.); raymondli@hku.hk (R.H.-W.L.); 2Department of Animal Science, Faculty of Animal Science and Export Agriculture, Uva Wellassa University, Badulla 90000, Sri Lanka; 3Shenzhen Key Laboratory of Fertility Regulation, The University of Hong Kong-Shenzhen Hospital, Shenzhen 518000, China

**Keywords:** LOPAC, trophoblastic spheroid attachment, endometrial receptivity, embryo implantation, Nemadipine-A

## Abstract

Current contraceptive methods interfere with folliculogenesis, fertilization, and embryo implantation by physical or hormonal approaches. Although hormonal contraceptive pills are effective in regulating egg formation, they are less effective in preventing embryo implantation. To explore the use of non-hormonal compounds that suppress embryo implantation, we established a high-throughput spheroid-endometrial epithelial cell co-culture assay to screen the Library of Pharmacologically Active Compounds (LOPAC) for compounds that affect trophoblastic spheroid (blastocyst surrogate) attachment onto endometrial epithelial Ishikawa cells. We identified 174 out of 1280 LOPAC that significantly suppressed BeWo spheroid attachment onto endometrial Ishikawa cells. Among the top 20 compounds, we found the one with the lowest cytotoxicity in Ishikawa cells, P11B5, which was later identified as Nemadipine-A. Nemadipine-A at 10 µM also suppressed BeWo spheroid attachment onto endometrial epithelial RL95-2 cells and primary human endometrial epithelial cells (hEECs) isolated from LH +7/8-day endometrial biopsies. Mice at 1.5 days post coitum (dpc) treated with a transcervical injection of 100 µg/kg Nemadipine-A or 500 µg/kg PRI-724 (control, Wnt-inhibitor), but not 10 µg/kg Nemadipine-A, suppressed embryo implantation compared with controls. The transcript expressions of endometrial receptivity markers, integrin αV (*ITGAV*) and mucin 1 (*MUC1*), but not β-catenin (*CTNNB1*), were significantly decreased at 2.5 dpc in the uterus of treated mice compared with controls. The reduction of embryo implantation by Nemadipine-A was likely mediated through suppressing endometrial receptivity molecules *ITGAV* and *MUC1*. Nemadipine-A is a potential novel non-hormonal compound for contraception.

## 1. Introduction

Embryo attachment onto the endometrium of the uterus is a critical step in establishing a pregnancy. However, the endometrium of the uterus is only receptive during a short period during the menstrual cycle [1]. In humans, following ovulation, progesterone secreted from the corpus luteum transforms the proliferative endometrium into a secretory state for embryo implantation on days 20–24 [2,3,4,5]. This short period is defined as the window of implantation (WOI) [2,6]. During the WOI, various proteins are expressed in the endometrium, including integrin αV (*ITGAV*) [7], β-catenin (*CTNNB1*) [8,9], and mucin 1 (*MUC1*), which also serve as endometrial receptivity markers [3]. Integrins are adhesion receptors that regulate cell–cell interactions, β-catenin acts as a transcriptional co-regulator of Wnt target genes, and mucin 1 modulates cell–cell interactions and cell–extracellular matrix interactions. These endometrial markers expressed in the luminal and glandular epithelium of the uterus are hormonally regulated [8,10,11,12]. High expression levels of integrin αVβ3 are associated with a successful pregnancy, whereas low expression levels are associated with unexplained infertility and endometriosis [13,14,15]. Compared with patients with normal levels of progesterone during the IVF cycle, some IVF patients with high levels of progesterone had suppressed *MUC1* and *CTNNB1* mRNA and protein expressions, which was associated with lower implantation rates [16].

Given that studies on human embryos are ethically challenging, in vitro co-cultures and mouse models are commonly used to investigate embryo implantation and factors that modulate implantation. Various in vitro spheroid–endometrial cell co-culture models have been established to investigate trophoblastic spheroid (blastocyst surrogate) attachment onto endometrial epithelial cells [17,18]. We have been using a trophoblastic–endometrial cells co-culture model to study the effects of endocrine-disrupting chemicals on implantation [19].

It is estimated that more than 25% of all pregnancies are unintended [20,21], and the use of emergency contraceptive (EC) pills is a common way to prevent unwanted pregnancies. Most EC pills contain high levels of hormones that inhibit ovulation, but they are less effective if ovulation has already occurred [22]. Furthermore, although EC pills nowadays have fewer side effects, some symptoms including headaches, dysmenorrhea, nausea, vomiting, and altered vaginal bleeding patterns are still reported [23]. Therefore, more effective novel non-hormonal contraceptive candidates await to be explored. In this study, we aimed to identify novel non-hormonal candidate compounds that can regulate endometrial receptivity and modulate embryo implantation. We attempted to identify small compounds from the Library of Pharmacologically Active Compounds (LOPAC) that can affect embryo implantation using a modified high-throughput co-culture assay reported previously (Ho et al., 2012) and to characterize the putative small compounds that have low cytotoxicity on spheroid attachment onto other human endometrial epithelial and primary endometrial epithelial cells. We also sought to confirm their effects using an in vivo mouse implantation model.

## 2. Results

### 2.1. Establishment and Validation of a High-Throughput In Vitro Model

We established a high-throughput in vitro co-culture assay to screen LOPAC for compounds that affect the attachment of trophoblastic spheroids on endometrial cells using a fluorescent labeling technique (Appendix A). Fluorescently labeled BeWo spheroids with a size of about 100 µm, similar in size to a human blastocyst, were generated by AggreWell. The optimal number of spheroids added per well was about 75, with almost no clumps formed by the spheroids (Appendix A). The fluorescence signal increased with the number of spheroids (from 0 to 100) seeded on the 96-well plate (Appendix A), and the R-square values ranged from 0.9619 to 0.8148. The fluorescence signal increased slightly after the 1 h co-culture and decreased after washing with PBS to remove unattached spheroids. No significant differences in the attachment rate were found with different numbers of seeded spheroids (25, 50, 75, and 100) (Appendix A).

We compared the attachment rates of spheroids determined by conventional manual counting and fluorescent readings. The attachment rates of BeWo spheroids onto Ishikawa cells were 90.5% and 51.0% by manual counting and by fluorescence reading, respectively (Figure 1A). The accuracy and reliability of the quantitative assessment of spheroid attachment by fluorescence signals were validated by treating the Ishikawa monolayer with PRI-724, a Wnt inhibitor (as a positive control), which significantly decreased the attachment of Ishikawa cells to 38.3% and 19.0% by manual counting and by fluorescence reading, respectively (Figure 1B). The fold changes in the attachment rate relative to the negative control were comparable in both detection methods (Figure 1C). Significant differences (*p* < 0.0001) were detected between the negative control and the treatment (PRI-724) group using both detection methods.

### 2.2. Primary Screening of LOPAC for Compounds That Suppress Spheroid Attachment

We performed a screening of the 1280 LOPAC compounds by repeated screening in duplicate samples using the modified high-throughput in vitro co-culture assay. The primary screening found 174 (13.6%) LOPAC compounds that suppressed the spheroid attachment rate (*p* < 0.05) compared with the negative and vehicle controls. Figure 1D shows the percentage changes in the attachment rates in descending order of the identified LOPAC compounds. A percentage change less than 0 in a treatment group represents a lower attachment rate than the negative control. For comparison, the PRI-724 treatment (positive control) induced the largest reduction in attachment rate (−62.7% change, *p* < 0.0001). The top 20 LOPAC compounds that induced the largest reduction in the spheroid attachment rate compared to the control (*p* < 0.05) were selected for the secondary screening (Table 1).

### 2.3. Secondary Screening of the Selected Compounds from LOPAC with Low Cytotoxicity

The cytotoxicity of the top 20 selected compounds was evaluated by XTT cell viability assay. Different concentrations of each selected compound (0.01, 0.1, 1, 10, and 100 μM) were used to treat Ishikawa cells for 24 h. The top 20 compounds (P1C7, P3A5, P3E6, P5D6, P8F6, P10B4, P11A8, P11B5, P11D5, P13A11, P13E7, P13G8, P14A7, P14E11, P14F4, P14G9, P15A9, P15E10, P16E5, and P16E7) and their LC_50_ values were determined (3.49, 0.09, 11.85, 6.91, 10.01, 0.02, 2.01, 17.37, 3.23, 0.35, 0.73, 1.93, 3.58, 0.33, 0.07, 0.51, 4.42, 0.11, 7.03, and 3.05 μM, respectively) (Figure 2A). Three out of the top 20 selected compounds, P3E6, P8F6, and P11B5, exhibited a LC_50_ value larger than the effective concentration (i.e., 10 μM), suggesting the suppressive effect was not mediated through cellular cytotoxicity. Compound P11B5 (Nemadipine-A) had the largest LC_50_ and was selected for further analysis. Detailed cytotoxicity assays and co-culture studies confirmed that P11B5 at 10 μM was able to suppress spheroid attachment onto Ishikawa cells and the LC_50_ was 41.25 μM (Figure 2B). We also found that P11B5 could suppress spheroid attachment onto other endometrial epithelial RL95-2 cells at 3 and 10 μM, and the LC_50_ was 41.87 μM (Figure 2C). 

### 2.4. Nemadipine-A Suppressed Attachment in Receptive Primary Human Endometrial Epithelial Cells (hEECs) and Ishikawa Cells with or without Steroid Hormones

We next studied the effect of Nemadipine-A at 10 µM on suppressing spheroid attachment onto primary hEECs under a hormone treatment of 500 pM β-estradiol and 50 nM progesterone. A total of nine human endometrial biopsies were collected on LH +7/8 days and used for hEECs isolation. The purity of the isolated hEECs and human endometrial stromal cells (hESCs) was determined by staining for epithelial (cytokeratin) and stromal (vimentin) cell markers, respectively (Figure 3A). The hEEC with purity >85% was used in the co-culture study. BeWo spheroids with homogeneous sizes (about 100 µm) were prepared by AggreWell (Figure 3B). The attached spheroids and total spheroids added onto the hEECs with or without Nemadipine-A treatment were measured to determine the attachment rate (Figure 3C). Nemadipine-A significantly suppressed (*p* < 0.05) the attachment of BeWo spheroids onto hEECs (6.9%) after 1 h of co-culture compared to the no treatment (17.5%) or vehicle (0.1% DMSO, 18.4%) controls. The spheroid attachment rate with the positive control PRI-724 at 10 μM (9.4%) was slightly higher than Nemadipine-A. To test if the presence of steroid hormones affects the attachment rate in the co-culture assay, we treated Ishikawa cells with or without steroid hormones before and during the co-culture assay. Both Nemadipine-A and PRI-724 significantly suppressed (*p* < 0.05 or *p* < 0.01 and *p* < 0.0001, respectively) the attachment of BeWo spheroids on Ishikawa cells with or without the E2 + P4 (estrogen: 10 nM; progesterone: 1 µM) when compared to the medium or vehicle (0.1% Ethanol or 0.1% DMSO) controls (Figure 3D,E).

### 2.5. In Vivo Effects of PBS, DMSO, PRI-724, and Nemadipine-A on Mouse Embryo Implantation

To study the in vivo effect of the small compound on embryo implantation, we adopted a non-invasive drug transfer technique to study the effect of LOPAC on mouse embryo implantation. It was determined that a volume of 3 μL was found to fully occupy the length of one uterine horn without any leakage at the vaginal opening in 7-week-old ICR mice (Appendix A). We therefore injected 3 μL PBS, DMSO, and PRI-724 (500 μg/kg) via transcervical catheter to one uterine horn at 1.5 dpc, with the contralateral uterine horn as the control. No significant effects were found on the number of implantation sites at 5.5 dpc between the control (no treatment) and PBS group (Appendix A), and between the control and DMSO group (Figure 4A). There was a significant (*p* < 0.001) suppressive effect on the number of implantation sites between the control and PRI-724 groups (*n* = 11–12, Figure 4A,B). As Nemadipine-A is a novel compound that has not been tested in vivo, we used similar concentrations based on a literature search and its solubility in DMSO diluted in PBS (maximum 5% DMSO). In line with the in vitro co-culture data, Nemadipine-A at 100 μg/kg significantly reduced the number of implantation sites on the treatment side compared with the contralateral control side at 5.5 dpc, whereas Nemadipine-A at 10 μg/kg had no suppressive effects (Figure 4A,B).

### 2.6. Nemadipine-A Suppressed Endometrial Receptivity Marker Expression in Mice

To delineate the molecular changes in uterine receptivity on embryo implantation induced by the Nemadipine-A treatment, we evaluated several endometrial receptivity markers including integrin αV (*ITGAV*), β-catenin (*CTNNB1*), and mucin 1 (*MUC1*) (Figure 4C). Nemadipine-A significantly downregulated the expression of *ITGAV* and *MUC1* (*p* < 0.05) but not *CTNNB1* transcripts when compared to the negative control.

## 3. Discussion

We screened 1280 LOPAC and identified Nemadipine-A as a possible candidate that suppresses BeWo spheroid attachment onto human endometrial epithelial cell lines and primary hEECs. The suppressive effect of Nemadipine-A on embryo implantation in vivo was confirmed in mice at 5.5 dpc after transcervical injection of the compound at 1.5 dpc. This suppressive effect on embryo implantation could be caused by the decreased expressions of endometrial receptivity markers *ITGAV* and *MUC1* in the treated uterus.

We first established an in vitro high-throughput co-culture model for screening LOPAC. Using AggreWell, we generated large numbers of homogeneous BeWo spheroids, which greatly reduced the time and effort in setting up the screening. BeWo spheroids were labeled with Calcein-AM, a live cell-permeant fluorescent dye, which facilitated the quantitative assessment of the attached and seeded spheroids. Although we found a slight increase in the fluorescence signals after co-culture, there was a decrease in the fluorescence signals after the washing step that removed unbound spheroids and dye leaking from the cells, which might explain why the attachment rates from manual counting and fluorescence reading methods differed slightly. Nevertheless, PRI-724 (Wnt-inhibitor) induced a similar fold change in the attachment rates determined by these two detection methods. The Wnt signaling molecules are expressed in the human endometrium [24], and Wnt activation favors embryo implantation [25]. PRI-724 specifically inhibits β-catenin and its coactivator CREB-binding protein [26,27], leading to a decrease in the receptivity of Ishikawa cells to BeWo spheroids. The emergency contraception mifepristone (10 μM) and ulipristal (4 μM) have been previously tested on both Ishikawa and trophoblastic JAr cells for 24 h, but only mifepristone slightly decreased the attachment of JAr spheroids onto Ishikawa cells from around 90% to 80% [28]. In our study, PRI-724 treatment in Ishikawa cells dramatically decreased the attachment rate.

LOPAC is a collection of small molecules that span a broad range of cell signaling pathways and are widely used in new drug discovery. The compounds cover all major target classes, such as kinase, proteases, G protein-coupled receptors (GPCRs), gene regulation, and neurotransmission. LOPAC has been used in antiviral and antifungal drug screening [29,30], neurotransmitter screening [31], drug discovery against hepatitis C virus and human immunodeficiency virus [32], and neurological diseases [33]. Among these publications, the common effective concentration and treatment time was 10 μM and 24 h [29,30,34,35]. In our primary screening of LOPAC, we used a concentration of 10 μM and a co-culture time of 24 h. Only compounds that significantly suppressed spheroid attachment with a LC_50_ higher than the effective concentration were selected for the secondary screening to determine their cytotoxicity. Among the top 20 candidates, Nemadipine-A had the largest LC_50_ value in Ishikawa and RL95-2 cells. Both Ishikawa and RL95-2 cells are receptive endometrial epithelial cell lines commonly used in implantation studies. They contain characteristic luminal and glandular epithelium and express various endometrial enzymes, adhesion molecules, proteins, and steroid receptors [36]. Moreover, human primary endometrial epithelial cells were used to confirm the results of the cell lines studied. The expressions of cytokeratins and vimentin are commonly used to verify the purity of the isolated human endometrial cells in cultures [37]. Primary hEECs are stained positively with cytokeratin and negatively with vimentin, whereas hESCs are positively stained with vimentin [28]. Our data suggested that Nemadipine-A could suppress spheroid attachment in both human cell lines and primary cells.

We optimized a transcervical transfer technique for our in vivo studies. In this study, intrauterine application of PRI-724 was used as the positive control to verify the successful transcervical transfer. In mice, the uterus is receptive at 3.5 dpc and blastocyst implantation occurs at midnight at 3.5 dpc [38]. To minimize the direct effects of the selected compounds on the embryo itself, we transferred the drug through the cervix at 1.5 dpc.

Nemadipine-A is an L-type calcium channel alpha-1 subunit antagonist. The L-type calcium channel (LTCC) is a high voltage-activated (HVA) calcium channel that is sensitive to 1,4-dihydropyridine (DHP) antagonists. The HVA calcium channel is a major type of voltage-gated calcium channel (VGCCs) [39], which are transmembrane ion channel proteins that guide calcium ions selectively through the cell membrane in response to membrane depolarization. Calcium-dependent inactivation was shown to involve LTCCs [40]. Nemadipine-A can bind to the alpha1 subunit of the LTCC to block the inward calcium current through a calcium-dependent inactivation mechanism. Accumulating evidence suggests that the endometrial calcium level has a critical role in establishing and maintaining pregnancy [41,42,43,44,45], but how changes in calcium levels affect endometrial gene expressions to regulate embryo implantation is not well defined and warrants further investigation.

To understand the molecular changes associated with Nemadipine-A treatment, the expression of endometrial receptivity markers was investigated. It was found that the expression of integrin αV (*ITGAV)* was decreased, suggesting Nemadipine-A treatment may downregulate the endometrial receptivity through integrin αV expression. In line with this, blocking integrin αVβ3 activity was reported to reduce the number of implantation sites in day four pregnant mice [46]. Similarly, the expression of β-catenin was found to be upregulated in the apical membrane of epithelial cells in the early morning on day five of pregnancy compared to early morning on day four or late evening on day five of pregnancy [47]. β-catenin knockout mice exhibited an aberrant decidualization response and gland formation [24]. Although we did not find any significant changes in *CTNNB1* mRNA expression in uterine epithelial cells after the treatment at 2.5 dpc compared to the control, a lower expression of *CTNNB1* transcript was found in the mouse endometrial epithelial cells. Changes in *CTNNB1* transcript expression on 3.5 or 4.5 dpc warrant investigation. Although mucin-1 acts as an anti-adhesion molecule during embryo attachment, it has differential expression patterns in mice and humans during pregnancy. In mice, the expression of *MUC1* is reduced during early pregnancy and disappears on days 4 to 5 when the embryo attaches to the epithelium [3,48]. Recent studies showed that the highly conserved cytoplasmic tail of mucin-1 can interact with β-catenin, suggesting mucin-1 has a potential role in regulating cell signaling [49,50]. In humans, the level of *MUC1* is decreased in the uterine flushing in patients suffering from recurrent spontaneous miscarriage [51]. Whether suppression of *MUC1* expression affects Wnt/β-catenin signaling and modulates endometrial receptivity in mice remains to be investigated.

In this study, we used the in vitro co-culture model and human cancer endometrial and trophoblast cell lines and cultured human primary endometrial cells, which may not represent the cellular response in vivo. Although the mouse is an ideal in vivo model for our experiment, the in vivo effects on the number of mouse implantation sites may not be translated into a human with a similar response. Also, the number of mice used is limited in this study.

Taken together, our results indicate that Nemadipine-A can suppress BeWo spheroid attachment onto pretreated human endometrial epithelial Ishikawa and RL95-2 cells and primary hEECs isolated from endometrial biopsies taken from LH +7/8 days. Transcervical transfer of Nemadipine-A reduced embryo implantation in mice partly through the downregulation of endometrial receptivity marker expressions.

## 4. Materials and Methods

### 4.1. Cell Culture

Human endometrial epithelial adenocarcinoma Ishikawa (EACC, Sigma-Aldrich, St. Louis, MO, USA) and RL95-2 cell lines (ATCC, St. Louis, MO, USA), and human trophoblastic choriocarcinoma BeWo cell line (ATCC, Manassas, Virginia, VA, USA) were cultured in plastic cell culture flasks in a 5% CO_2_ humidified environment at 37 °C. Ishikawa and RL95-2 cells were maintained in minimum essential medium (MEM, Sigma, St. Louis, MO, USA), whereas BeWo cells were maintained in Dulbecco’s modified Eagle’s medium/nutrient mixture F-12 Ham (DMEM/F12, Sigma, St. Louis, MO, USA). All media were supplemented with 10% fetal bovine serum (FBS, Invitrogen, Carlsbad, CA, USA) and 1% L-glutamine (Gibco, Life Technologies, Carlsbad, CA, USA), and 1% penicillin/streptomycin (Life Technologies, Carlsbad, CA, USA). Confluent cells were dispersed by trypsin/EDTA (Life Technologies, Carlsbad, CA, USA) before subculture.

### 4.2. Treatment of Endometrial Epithelial Cells with Steroid Hormones and LOPAC

Ishikawa cells were starved with 5% charcoal/dextran-stripped FBS (cs-FBS, Hyclone, Logan, UT, USA) in phenol red-free MEM for 48 h before treating with or without estrogen (10 nM, Sigma, St. Louis, MO, USA) and progesterone (1 µM, Sigma, St. Louis, MO, USA) for 24 h. Nemadipine-A from LOPAC (Sigma Aldrich, St. Louis, MO, USA) and PRI-724 (Wnt pathway inhibitor, ab229168, Abcam, Waltham, MA, USA) were used to treat Ishikawa cells with or without the above hormones for 24 h before the high-throughput attachment assay.

### 4.3. Isolation and Culture of Primary hEECs

Primary hEECs were collected by endometrial biopsy at the mid-luteal phase (LH +7/8 days) from females with regular menstrual cycles [28]. The hEECs were isolated as previously described [52,53] and then cultured in MEM/F12 medium containing 10% cs-FBS, 1% P/S, 1% L-Glu, 500 pM β-estradiol (Sigma, St. Louis, MO, USA) and 50 nM progesterone (Sigma, St. Louis, MO, USA) at the same hormonal levels as in the mid-luteal phase. Written informed consent was obtained from patients before collecting samples. The study protocol was approved by the Institutional Review Board, The University of Hong Kong/Hospital Authority Hong Kong West Cluster (IRB: UW 17-458).

### 4.4. Spheroid Attachment Assay 

A high-throughput spheroid attachment assay was performed as previously described [54] with some modifications (Appendix A). Endometrial epithelial cells in a culture flask were trypsinized and seeded at $2.3\times 10^{4}$ cells per well onto a 96-well plate (Costar, Washington, DC, USA) in the complete MEM medium. The cells were allowed to form a confluent monolayer over 24 h. The medium was changed to phenol red-free MEM with or without supplementation of compounds from LOPAC. After 24 h, the medium was changed with fresh phenol red-free MEM without the compounds before the co-culture assay. Each run included control wells with background references (medium and cells), negative control (no treatment), vehicle control (0.1% DMSO), and positive control (10 μM PRI-724) [26]. Controls were run in quadruplicate and treatment wells (10 µΜ LOPAC) were run in duplicate. Two runs were carried out in each experiment.

For the generation of spheroids, BeWo cells were dissociated into single cells and seeded at $1.2\times 10^{5}\;$cells per well in an AggreWell 400 (STEMCELL Technologies, Vancouver, BC, Canada). After brief centrifugation, cells were allowed to aggregate for 24 h in a 5% CO_2_ humidified environment at 37 °C. After BeWo cells had formed homogeneous spheroids, they were labeled with fluorescent Calcein-AM dye (0.5 μg/mL, Sigma Aldrich, St. Louis, MO, USA) added to the medium and incubated for 30 min. The BeWo spheroids were washed and diluted in a reservoir (Costar, Washington, DC, USA). About 75 labeled spheroids (about one spheroid/μL) were transferred onto the epithelial monolayer using a multi-pipette (Eppendorf 300, Merck, NJ, USA) for the co-culture assay. The spheroid-epithelial cells co-culture was incubated for 1 h in a 5% CO_2_ humidified environment at 37 °C. The number of seeded BeWo spheroids was determined by measuring the fluorescence signal using an automated plate reader (Ex/Em = 485 nm/535 nm) (Tecan, Männedorf, Switzerland). The unattached spheroids in each well were washed away with 200 μL of phosphate-buffered saline (PBS) and the fluorescence signal of the attached spheroids was measured again. The attachment rate was calculated as the fluorescence signal of the attached spheroids divided by the fluorescence signal of the seeded spheroids as a percentage (the change in the attachment rate = 100% × (the attachment rate of the treatment group − the mean attachment rate of the negative control)/the mean attachment rate of the negative control).

For the hEEC co-culture assay, hEECs were cultured for 3 to 5 days until 90% confluency in 5-µL droplets. The cells were treated with or without 0.1% DMSO (vehicle control), PRI-724 (10 µΜ, positive control), and Nemadipine-A (10 µΜ, LOPAC) for 24 h. BeWo spheroids, prepared as above, at around 23 to 30 spheroids per droplet were transferred onto the hEEC monolayer using a fine Pasteur glass pipette (Sigma Aldrich, St. Louis, MO, USA) with an internal diameter of 200 to 250 µm and then incubated for 1 h at 37 °C. After co-culture, the plate was gently patted before the unattached spheroids were aspirated using a glass pipette. The effect of the treatments on spheroid attachment was evaluated by calculating the number of attached spheroids compared to the total added as a percentage.

### 4.5. Cell Viability Assay

The effects of the LOPAC compounds on the viability of Ishikawa cells were studied using the CyQuant XTT Cell Viability Assay Kit (Invitrogen, Carlsbad, CA, USA). Ishikawa cells were seeded at $5\times 10^{3}\;$cells per well in a 96-well plate containing 100 µL MEM medium for 24 h. The medium was changed to a fresh medium containing LOPAC (0.01–100 μM) and incubated for a further 24 h. XTT Reagent was then added and incubated for a further 4 h. The absorbance at 450 nm and 660 nm was measured and the specific absorbance of the sample was calculated (Specific Absorbance = A_450nm_Test
− A_450nm_Blank − A_660nm_Test). The percentage of cell viability was calculated relative to the control (DMSO). The LC_50_ values were also calculated.

### 4.6. Immunofluorescence Staining

Primary endometrial epithelial cells and stromal cells were seeded onto 48-well plates and grown until 90% confluency. The cells were washed with 1X PBS before being fixed with 10% neutral formalin for 20 min at room temperature. Cells were permeated with 0.1% Triton for 15 min and non-specific binding sites were blocked with 10% normal goat serum for 1 h. After washing twice with PBST, cells were incubated with primary antibodies against cytokeratin (AE1/AE3, 1:100, IR053, DAKO, Glostrup, Denmark) and vimentin (1:100, M0725, DAKO, Glostrup, Denmark) in the blocking medium at 4 °C overnight. The cells were washed with PBST four times before incubating with a secondary goat anti-mouse antibody with Alexa Fluor 488 (1:500, Abcam, MA, USA) in a blocking solution at room temperature for 1 h. After washing, the nuclei were stained with DAPI (20 µg/mL, Invitrogen, Carlsbad, CA, USA) in PBST for 3 min. Fluorescence images were captured under a fluorescence microscope (Nikon, Tokyo, Japan).

### 4.7. Murine Model and Transcervical Transfer of LOPAC

Adult female ICR mice (6–7 weeks old) were obtained from the AAALAC International accredited Center for Comparative Medicine Research (CCMR), The University of Hong Kong. Mice were housed in the facility under climate-controlled conditions (21 ± 2 °C) in a 12-h light/12-h dark cycle. All procedures followed the protocol approved by the Committee on the Use of Live Animals in Teaching and Research, The University of Hong Kong (Reference number: CULATR 4966-19). Briefly, female ICR mice were mated with stud male mice, and the detection of the vaginal plug was counted as 0.5 days post coitum (dpc). At 1.5 dpc, 3 μL of 1X PBS, 5% DMSO in 1X PBS, and 500 μg/kg PRI-724 (dissolved in 5% DMSO, Sigma, St. Louis, MO, USA) or Nemadipine-A (100 and 10 μg/kg, Sigma, St. Louis, MO, USA) was transferred to the right uterine horn of anesthetized female mice by a transcervical transfer device [55].

### 4.8. RNA Extraction and Quantitative Reverse Transcription-Polymerase Chain Reaction (qRT—PCR)

To study the expression of endometrial receptivity markers, the endometrial epithelium was scratched from the dissected uterine horn at 2.5 dpc under a microscope (Nikon). Total RNA was extracted using a mirVana PARIS RNA extraction kit (AM1556, Life Technologies, Carlsbad, CA, USA). Briefly, the scratched tissue samples were homogenized, and total RNA was eluted with the Elution Solution. The concentration and quality of isolated RNA were measured using a Nanodrop 200c (Thermo Fisher, Waltham, MA, USA). Only RNA samples with a 260/280 nm ratio above 1.8 were used for the subsequent experiments and were stored at −80 °C until use.

The isolated RNA was reverse-transcribed (RT) using the TaqMan Reverse Transcription Reagents (N8080234, Applied Biosystems, Waltham, MA, USA) following the manufacturer’s protocol. Each 20 μL RT reaction volume contained 700 ng of total RNA in nuclease-free water. Complementary DNA (cDNA) was synthesized (thermal cycling conditions: 25 °C for 10 min, 37 °C for 60 min, and 95 °C for 5 min) and stored at −20 °C until use. The qPCR was carried out in triplicate on 10 μL reaction volumes containing TaqMan Universal PCR Master Mix (Life Technologies, Carlsbad, CA, USA) using the QuantStudio 5 Real-Time PCR System (Applied Biosystems, Waltham, MA, USA). TaqMan probes for *ITGAV* (Mm00434486_m1), *CTNNB1* (Mm00483039_m1), *MUC1* (Mm00449604_m1), and 18S VIC reporter (4319413E, Applied Biosystems, Waltham, MA, USA) were used. The expression of mRNA was quantified by the ΔΔCT method [56] and the results were expressed as fold changes (2-ΔΔCT) relative to the negative control (1.5 dpc samples).

### 4.9. Statistical Analysis

Changes in the attachment rate were plotted as the mean in R statistical software package (version3.0.1, Boston, MA, USA). Statistical analyses were performed using GraphPad Prism 7 (GraphPad Software, La Jolla, CA, USA), and values were presented as the mean ± SD. The *p*-values of the experiments were determined by the Mann–Whitney U test, Wilcoxon matched–pairs signed–rank test, or Kruskal–Wallis test, as appropriate. A *p*-value of less than 0.05 was considered statistically significant.

## Figures and Tables

**Figure 1 ijms-23-05073-f001:**
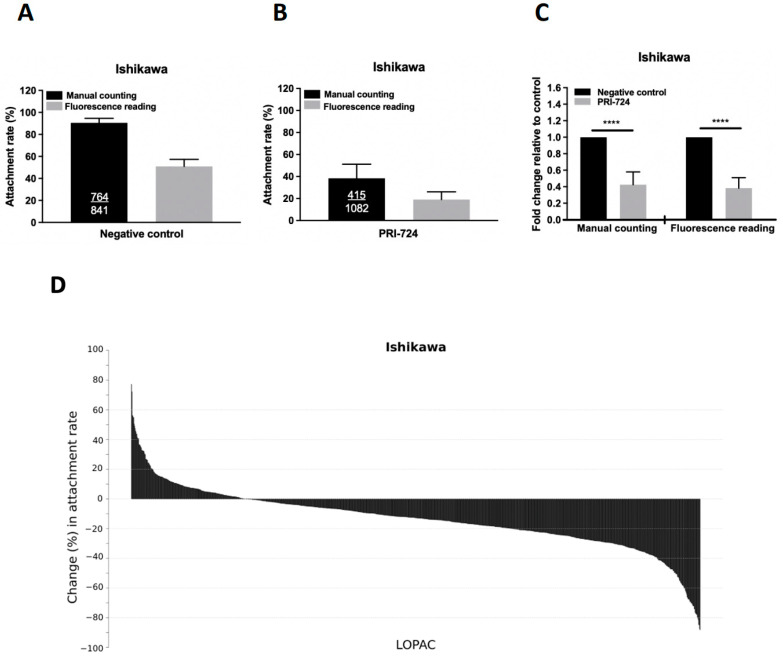
Establishment of the high-throughput attachment assay and primary screening of the LOPAC library. Comparison of the attachment rates of BeWo spheroids onto Ishikawa cells with manual counting (black bar) and fluorescence quantitation (grey bar) in (**A**) negative control, (**B**) positive control with PRI-724, and (**C**) combined results as fold changes in Ishikawa cells with (grey bar) or without (black bar) treatment of PRI-724 by manual counting and fluorescence reading. Ishikawa cells were treated with PRI-724 for 24 h and the number of attached spheroids was manually counted or measured by reading the fluorescence signals. The number of attached spheroids over the number of total spheroids added is shown in the black bars. The attachment rate (percentage) is represented as the mean ± SD and each experiment was repeated three times in triplicate. **** denotes at *p* < 0.0001 in the Mann–Whitney U test. (**D**) The percentage changes of BeWo spheroid attachment rates on Ishikawa cells treated with 10 μM LOPAC are presented from the highest to the lowest.

**Figure 2 ijms-23-05073-f002:**
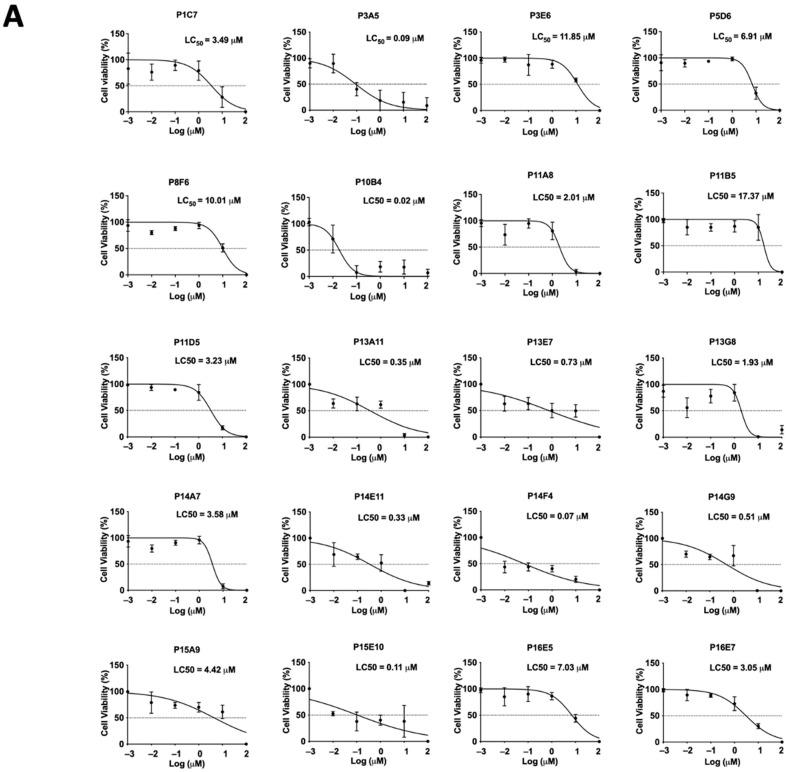
Cell viability assay in selected LOPAC and high-throughput attachment assay in Nemadipine-A (P11B5). (**A**) Ishikawa cells were treated with or without different concentrations (0.01, 0.1, 1, 10, and 100 μM) of the selected LOPAC for 24 h. Cell viability was determined by XTT assay and values were compared to vehicle control (0.1% DMSO) and represented as percentages (*n* = 3). The spheroid attachment in the established high-throughput co-culture model and cytotoxic effects of Nemadipine-A (P11B5, 0.01–100 μM) were determined in human receptive endometrial epithelial, (**B**) Ishikawa, and (**C**) RL95-2 cells. *, **, and **** denote *p* < 0.05, <0.01, and <0.0001 in the Mann–Whitney U test or Kruskal–Wallis test followed by Dunn’s multiple comparisons test compared to medium or 0.1% DMSO control. PRI-724 at 10 μM was used as a positive control in the co-culture assay.

**Figure 3 ijms-23-05073-f003:**
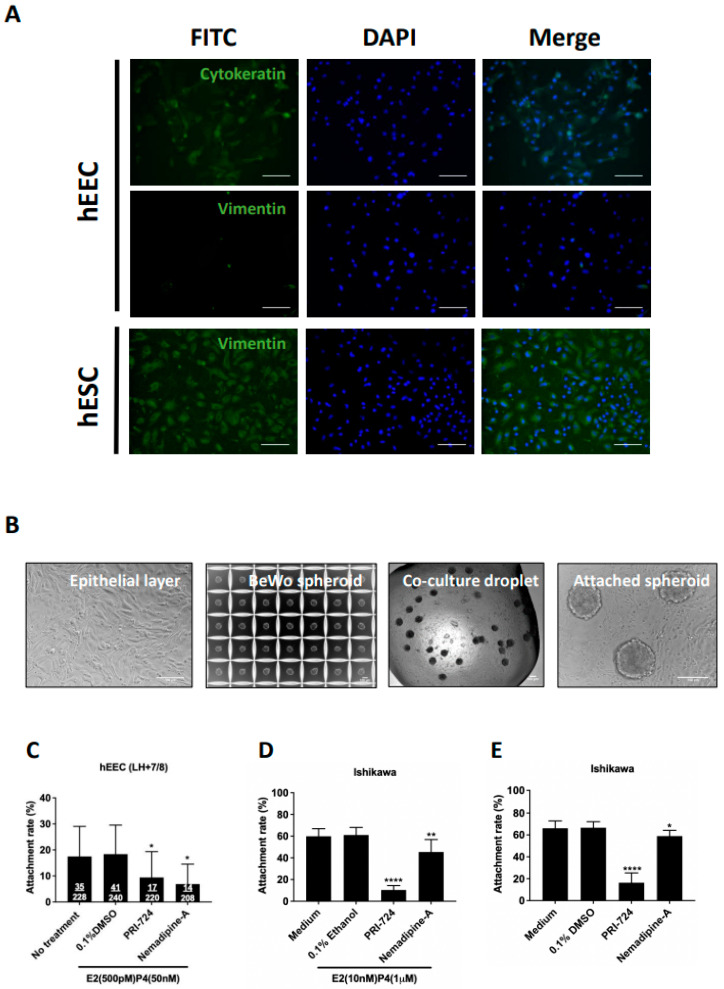
The effect of Nemadipine-A on BeWo spheroid attachment on primary human endometrial epithelial cells (hEECs) and Ishikawa cells with or without steroid hormone treatment. (**A**) The purity of the isolated primary hEECs and hESCs were determined by cytokeratin (epithelial cell marker, green) and vimentin (stromal cell marker, green) staining, respectively. More than 85% of the hEECs were positive for cytokeratin, and more than 90% of the isolated hESCs were positive for vimentin. All cells were counter-stained with DAPI (nucleus staining, blue). (**B**) Representative micrographs of the isolated primary hEEC monolayer, BeWo spheroids generated in AggreWell, co-culture droplet, and attached spheroids on primary hEEC monolayer. (**C**) Effect of Nemadipine-A on the attachment rate of BeWo spheroids on primary hEECs isolated from endometrium aspirate at LH +7/8 days and maintained in physiological 500 pM β-estradiol and 50 nM progesterone (*n* = 9). Effect of Nemadipine-A on the attachment rate of BeWo spheroids onto Ishikawa cells with (**D**) or without (**E**) the 48-h E2 + P4 (E2: 10 nM; P4: 1 µM) treatment (*n* = 3). Mean ± SD, *, **, and **** denote *p* < 0.05, <0.01, and <0.0001 compared to medium or vehicle (DMSO or ethanol) controls in Mann–Whitney U test. Scale bars = 100 μm.

**Figure 4 ijms-23-05073-f004:**
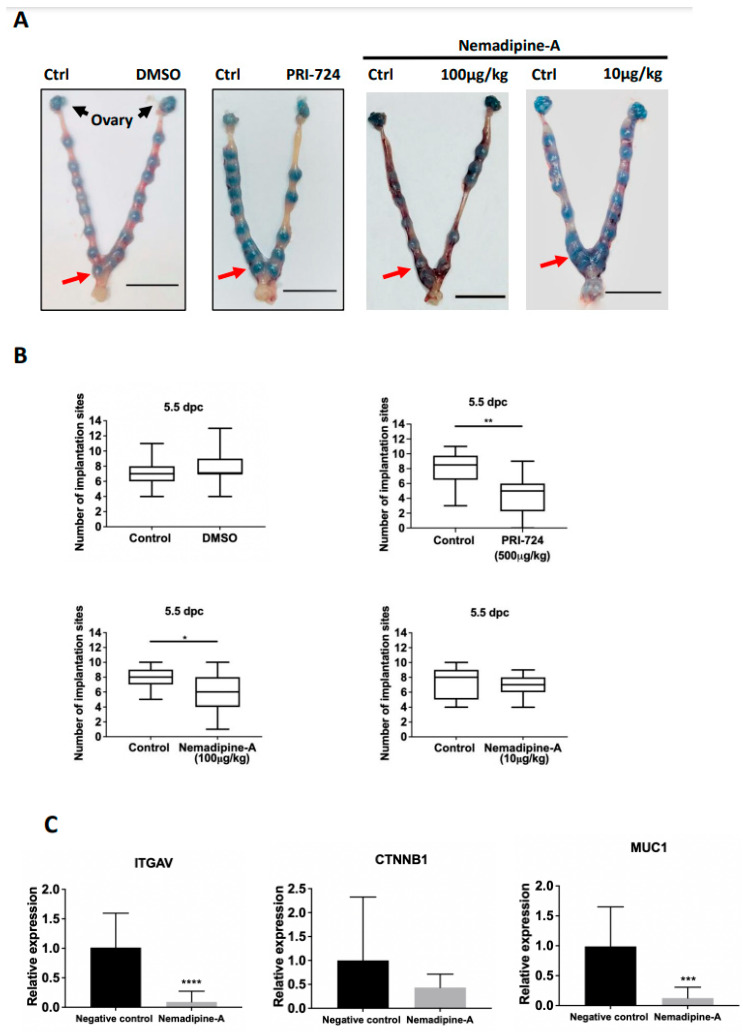
The effect of Nemadipine-A on embryo implantation in mice and changes in the expression of endometrial receptivity genes. (**A**) Representative images of mice uteri at 5.5 dpc after injection with 3 μL DMSO, PRI-724, and Nemadipine-A (10 and 100 μg/kg) at 1.5 dpc. The number of implantation sites (red arrow) at 5.5 dpc was counted via tail vein injection of Chicago blue (Scale bars = 1 cm). (**B**) The number of implantation sites on the Nemadipine-A-treated and control sides at 5.5 dpc was measured. Data are presented as median (2.5–9.75th percentile). * and ** denote *p* < 0.05 and <0.01 in Wilcoxon matched-pairs signed-rank test compared to the control (*n* = 11–12). (**C**) Quantitative PCR analysis of the expression of *ITGAV*, *CTNNB1*, and *MUC1* transcripts in endometrial epithelium from 2.5 dpc ICR mice after transcervical transfer of Nemadipine-A (100 μg/kg) for 24 h. Data are represented as mean ± SD. *** and **** denote *p* < 0.001 and <0.0001 in the Mann–Whitney U test compared to the control (*n* = 3–4).

**Table 1 ijms-23-05073-t001:** The top 20 LOPAC (compound ID) with the highest changes in attachment rate.

Molecule ID	Change (%) in	*p*-Value	LC_50_ (µM)
	Attachment Rate (at 10 μM)		
P14A7	−88.3	0.0142	3.58
P3E6	−87.5	0.0286	11.85
P14F4	−84.8	0.004	0.07
P11A8	−84.7	0.004	2.01
P15A9	−81.3	0.0095	4.42
P13A11	−80.3	0.004	0.35
P1C7	−79.8	0.0286	3.49
P14G9	−78.1	0.0402	0.51
P13G8	−77.6	0.004	1.93
P15E10	−77.3	0.004	0.11
P11D5	−77.1	0.004	3.23
P3A5	−76.5	0.0286	0.09
P8F6	−74.4	0.0286	10.01
P11B5	−74.0	0.004	17.37
P10B4	−72.1	0.0286	0.02
P13E7	−72.1	0.004	0.73
P16E7	−71.9	0.004	3.05
P16E5	−71.3	0.004	7.03
P14E11	−70.5	0.004	0.33
P5D6	−55.2	0.0286	6.91

Half-lethal concentration (LC_50_), *p* < 0.05 (Mann−Whitney U test).

## Data Availability

The data presented in this study are available on request from the corresponding author.

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
