# Peer review of "High-Throughput In Vitro Screening Identified Nemadipine as a Novel Suppressor of Embryo Implantation"

_ijms, 2022, doi:10.3390/ijms23095073_

Round 1
Reviewer 1 Report
The manuscript by X. Chen et. al., titled " High-throuput in vitro screening ......." is a sound well written manuscript describing phase I experiment for the identification and evaluation of Nemadipine-A as an implantation suppressor in experimental animals and a potential contraceptive in humans. The authors have applied a high-throughput approach to screen for the appropriate compounds in the LOPAC library and have also tested the effercts of the selected compound on three relevant endometrial receptivity marker genes.
Author Response
We thank the reviewer's positive feedback on our manuscript.
Reviewer 2 Report
Authors set up a high-throughput in vitro co-culture assay to screen LOPAC for compounds that affect the attachment of trophoblastic spheroids on endometrial cells. They found one compound named Nemadipine-A which can interrupted embryos implantation. The mechanism of Nemadipine-A may be suppressed by expressions of endometrial receptivity marker (ITGAV and MUC1, but not CTNNB1) transcripts, were significantly decreased at 2.5 dpc in the uterus of treated mice compared with controls. There are some questions of this manuscript as following list.
- Authors designed transferred different reagent or compounds in two sites of mouse uterus. The true effect of Nemadipine-A on embryo implantation in mice should not be effect by another reagent.How authors avoid reagents cross-over from one site to another site?
- Authors analyzed the effect of Nemadipine-A on embryo implantation in mice and changes in the expression of endometrial receptivity genes. Although there was statistic significant between control group and ITGAV or MUC1 genes.
The standard derivation of gene expression showed high variation in figure 4 C) in 3 to 4 repeated tests. Was there any variant of animal? Authors need to discuss this observation. Otherwise, the suppression effect of Nemadipine-A seemed not stable on individual differences.
- Figure 2A was too small to read. Is it necessary to show all figures of 2A?
Author Response
Q1. Authors designed transferred different reagent or compounds in two sites of mouse uterus. The true effect of Nemadipine-A on embryo implantation in mice should not be effect by another reagent. How authors avoid reagents cross-over from one site to another site?
A1. We have performed a preliminary study using blue dye to optimize the volume of chemicals in the transcervical transfer study (Suppl figure S2A). We believe leakage of drugs through the cervical os is minimal in our study.
Q2. Authors analyzed the effect of Nemadipine-A on embryo implantation in mice and changes in the expression of endometrial receptivity genes. Although there was statistic significant between control group and ITGAV or MUC1 genes. The standard derivation of gene expression showed high variation in figure 4 C) in 3 to 4 repeated tests. Was there any variant of animal? Authors need to discuss this observation. Otherwise, the suppression effect of Nemadipine-A seemed not stable on individual differences.
A2. Thank you for your comment concerning the large variation in the relative gene expression levels. The relative changes are almost 10-fold in ITGAV and MUC1, but not CTNNB1. We have performed 3-4 mice in this part of the study and agreed that the number of animals used is limited, yet is significantly different between treatment groups. We address the reviewer's concern by adding "Also, the number of mice used is limited in this study." in the second last paragraph of the discussion section.
Q3. Figure 2A was too small to read. Is it necessary to show all figures of 2A?
A3. Figure 2A provided information on the LC50 of the 20 selected drugs (Table 1) and their effect on cell viability at different concentrations. I hope the reviewer will agree this piece of information is useful and important to the reader.